# Search-Guided, Lightly-Supervised Training of Structured Prediction Energy Networks

**Amirmohammad Rooshenas, Dongxu Zhang, Gopal Sharma, and Andrew McCallum**
College of Information of Computer Sciences
University of Massachusetts Amherst
Amherst, MA 01003
{pedram,dongxuzhang,gopalsharma,mccallum}@cs.umass.edu

## Abstract

In structured output prediction tasks, labeling ground-truth training output is often expensive. However, for many tasks, even when the true output is unknown, we can evaluate predictions using a scalar reward function, which may be easily assembled from human knowledge or non-differentiable pipelines. But searching through the entire output space to find the best output with respect to this reward function is typically intractable. In this paper, we instead use efficient truncated randomized search in this reward function to train structured prediction energy networks (SPENs), which provide efficient test-time inference using gradient-based search on a smooth, learned representation of the score landscape, and have previously yielded state-of-the-art results in structured prediction. In particular, this truncated randomized search in the reward function yields previously unknown local improvements, providing effective supervision to SPENs, avoiding their traditional need for labeled training data.

## 1 Introduction

Structured output prediction tasks are common in computer vision, natural language processing, robotics, and computational biology. The goal is to find a function from an input vector $\mathbf{x}$ to multiple coordinated output variables $\mathbf{y}$. For example, such coordination can represent constrained structures, such as natural language parse trees, foreground-background pixel maps in images, or intertwined binary labels in multi-label classification.

Structured prediction energy networks (SPENs) (Belanger & McCallum, 2016) are a type of energy-based model (LeCun et al, 2006) in which inference is done by gradient descent. SPENs learn an energy landscape $E(\mathbf{x}, \mathbf{y})$ on pairs of input $\mathbf{x}$ and structured outputs $\mathbf{y}$. In a successfully trained SPEN, an input $\mathbf{x}$ yields an energy landscape over structured outputs such that the lowest energy occurs at the target structured output $\mathbf{y}^*$. Therefore, we can infer the target output by finding the minimum of energy function $E$ conditioned on input $\mathbf{x}$: $\mathbf{y}^* = \mathrm{argmin}_{\mathbf{y}} E(\mathbf{x}, \mathbf{y})$.

Traditional supervised training of SPENs requires knowledge of the target structured output in order to learn the energy landscape, however such labeled examples are expensive to collect in many tasks, which suggests the use of other cheaply acquirable supervision. For example, Mann and McCallum (2010) use labeled features instead of labeled output, or Ganchev et al. (2010) use constraints on posterior distributions of output variables, however both directly add constraints as features, requiring the constraints to be decomposable and also be compatible with the underlying model's factorization to avoid intractable inference.

Alternatively, scalar reward functions are another widely used source of supervision, mostly in reinforcement learning (RL), where the environment evaluates a sequence of actions with a scalar

reward value. RL has been used for direct-loss minimization in sequence labeling, where the reward function is the task-loss between a predicted output and target output (Bahdanau et al., 2017; Maes et al., 2009), or where it is the result of evaluating a non-differentiable pipeline over the predicted output (Sharma et al., 2018). In these settings, the reward function is often non-differentiable or has low-quality continuous relaxation (or surrogate) making end-to-end training inaccurate with respect to the task-loss.

Interestingly, we can also rely on easily accessible human domain-knowledge to develop such reward functions, as one can easily express output constraints to evaluate structured outputs (e.g., predicted outputs get penalized if they violate the constraints). For example, in dependency parsing each sentence should have a verb, and thus parse outputs without a verb can be assigned a low score.

More recently, Rooshenas et al. (2018) introduce a method to use such reward functions to supervise the training of SPENs by leveraging rank-based training and SampleRank (Rohanimanesh et al., 2011). Rank-based training shapes the energy landscape such that the energy ranking of alternative **y** pairs are consistent with their score ranking from the reward function. The key question is how to sample the pairs of **y**s for ranking. We don't want to train on all pairs, because we will waste energy network representational capacity on ranking many unimportant pairs irrelevant to inference; (nor could we tractably train on all pairs if we wanted to). We do, however, want to train on pairs that are in regions of output space that are misleading for gradient-based inference when it traverses the energy landscape to return the target. Previous methods have sampled pairs guided by the thus-far-learned energy function, but the flawed, preliminarily-trained energy function is a weak guide on its own. Moreover, reward functions often include many wide plateaus containing most of the sample pairs, especially at early stages of training, thus not providing any supervision signal.

In this paper we present a new method providing efficient, light-supervision of SPENs with margin-based training. We describe a new method of obtaining training pairs using a combination of the model's energy function and the reward function. In particular, at training time we run the test-time energy-gradient inference procedure to obtain the first element of the pair; then we obtain the second element using randomized search driven by the reward function to find a local true improvement over the first. Using this search-guided approach we have successfully performed lightly-supervised training of SPENs with reward functions and improved accuracy over previous state-of-the-art baselines.

## 2   Structured Prediction Energy Networks

A SPEN parametrizes the energy function $E_{\mathbf{w}}(\mathbf{y}, \mathbf{x})$ using deep neural networks over input $\mathbf{x}$ and output variables $\mathbf{y}$, where $\mathbf{w}$ denotes the parameters of deep neural networks. SPENs rely on parameter learning for finding the correlation among variables, which is significantly more efficient than learning the structure of factor graphs. One can still add task-specific bias to the learned structure by designing the general shape of the energy function. For example, Belanger and McCallum (2016) separate the energy function into global and local terms. The role of the local terms is to capture the dependency among input $\mathbf{x}$ and each individual output variable $y_i$, while the global term aims to capture long-range dependencies among output variables. Gygli et al. (2017) define a convolutional neural network over joint input and output.

Inference in SPENs is defined as finding $\operatorname{argmin}_{\mathbf{y} \in \mathcal{Y}} E_{\mathbf{w}}(\mathbf{y}, \mathbf{x})$ for given input $\mathbf{x}$. Structured outputs are represented using discrete variables, however, which makes inference an NP-hard combinatorial optimization problem. SPENs achieve efficient approximate inference by relaxing each discrete variable to a probability simplex over the possible outcome of that variable. In this relaxation, the vertices of a simplex represent the exact values. The simplex relaxation reduces the combinatorial optimization to a continuous constrained optimization that can be optimized numerically using either projected gradient-descent or exponentiated gradient-descent, both of which return a valid probability distribution for each variable after every update iteration.

Practically, we found that exponentiated gradient-descent, with updates of the form $y_i^{t+1} = \frac{1}{Z_i^t} y_i^t \exp(-\eta \frac{\partial E}{\partial y_i})$ (where $Z_i^t$ is the partition function of the unnormalized distribution over the values of variable $i$ at iteration $t$) improves the performance of inference regarding convergence and finds better outputs. This is in agreement with similar results reported by Belanger et al. (2017) and Hoang et al. (2017). Exponentiated gradient descent is equivalent to defining $y_i = \text{Softmax}(I_i)$,

where $I_i$ is the logits corresponding to variable $y_i$, and taking gradient descent in $I_i$, but with gradients respect to $y_i$ (Kivinen & Warmuth, 1997): $I_i^{t+1} = I_i^t - \eta \frac{\partial E}{\partial y_i}$.

Multiple algorithms have been introduced for training SPENs, including structural SVM (Belanger & McCallum, 2016), value-matching (Gygli et al., 2017), end-to-end training (Belanger, 2017), and rank-based training (Rooshenas et al., 2018). Given an input, structural SVM training requires the energy of the target structured output to be lower than the energy of the loss-augmented predicted output. Value-matching (Gygli et al., 2017), on the other hand, matches the value of energy for adversarially selected structured outputs and annotated target structured outputs (thus strongly-supervised, not lightly-supervised) with their task-loss values. Therefore, given a successfully trained energy function, inference would return the structured output that minimizes the task-loss. End-to-end training (Belanger et al., 2017) directly minimizes a differentiable surrogate task-loss between predicted and target structured outputs. Finally, rank-based training shapes the energy landscape such that the structured outputs have the same ranking in the energy function and a given reward function.

While structural SVM, value-matching, and end-to-end training require annotated target structured outputs, rank-based training can be used in domains where we have only light supervision in the form of reward function $R(\mathbf{x}, \mathbf{y})$ (which evaluates input $\mathbf{x}$ and predicted structured output $\mathbf{y}$ to a scalar reward value). Rank-based training collects training pairs from a gradient-descent trajectory on energy function. However, these training trajectories may not lead to relevant pairwise rank violations (informative constraints that are necessary for training (Huang et al., 2012)) if the current model does not navigate to regions with high reward. This problem is more prevalent if the reward function has plateaus over a considerable number of possible outputs—for example, when the violation of strong constraints results in constant values that conceal partial rewards. These plateaus happen in domains where the structured output is a set of instructions such as a SQL query, and the reward function evaluates the structured outputs based on their execution results.

This paper introduces a new search-guided training method for SPENs that addresses the above problem, while preserving the ability to learn from light supervision. As described in detail below, in our method the gathering of informative training pairs is guided not only by gradient descent on the thus-far-learned energy function, but augmented by truncated randomized search informed by the reward function, discovering places where reward training signal disagrees with the learned energy function.

## 3   Search-Guided Training

Search-guided training of SPENs relies on a randomized search procedure $S(\mathbf{x}, \mathbf{y}_s)$ which takes the input $\mathbf{x}$ and starting point $\mathbf{y}_s$ and returns a successor point $\mathbf{y}_n$ such that

$$R(\mathbf{x}, \mathbf{y}_n) > R(\mathbf{x}, \mathbf{y}_s) + \delta, \tag{1}$$

where $\delta > 0$ is the search margin that controls the complexity of the search operator. For large $\delta$, the search operator requires more exploration to satisfy eq. 1 while the returned successor point $\mathbf{y}_n$ is closer to the true output that maximizes the reward function, thus providing a stronger supervision signal. Smaller values of $\delta$, on the other hand, require less exploration, but provide weaker supervision signal; (see Appendix B for a comparison on reward margin values). Of course, many randomized search procedures are possible—simple and complex.

In the experiments of this paper we find that a simple randomized search works well: we start from the gradient-descent inference output, iteratively select a random output variable, uniformly sample a new state for the selected variable; if the reward increases more than the margin, return the new sample; if the reward increases less than the margin, similarly change an additional randomly selected variable; if the reward decreases, undo the change, and begin the sampling again. (If readily available, domain knowledge could be injected into the search to better explore the reward function; this is the target of future work.) We truncate the randomized search by bounding the number of times that it can query the reward function to evaluate structured outputs for each input $\mathbf{x}$ at every training step. As a result, the search procedure may not be able to find a local improvement (this also may happen if $\mathbf{y}_s$ is already near-optimal), in which case we simply ignore that training example in the current training iteration. Note that the next time that we visit an ignored example, the inference procedure may provide a better starting point or truncated randomized search may find a local improvement. In practice we observe that, as training continues, the truncated randomized search finds local improvements for every training point (see Appendix C).

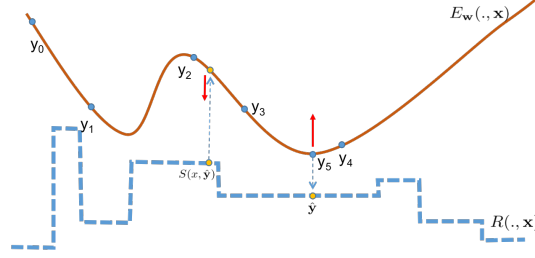

Figure 1: Search-guided training: the solid and dashed lines show a schematic landscape of energy and reward functions, respectively. The blue circles indexed by $\mathbf{y}_i$ represent the gradient-descent inference trajectory with five iterations over the energy function. Dashed arrows represent the mapping between the energy and reward functions, while the solid arrows show the direction of updates.

Intuitively, we are sampling $\hat{\mathbf{y}}$ from the energy function $E(\mathbf{y}, \mathbf{x})$ by adding Gaussian noise (with the standard deviation of $\sigma$) to the gradient descent on logits: $I_i^{t+1} = I_i^t - \eta \frac{\partial E}{\partial y_i} + \mathcal{N}(0, \sigma)$, which is similar to using Langevin dynamics for sampling from a Boltzmann distribution.

Via the search procedure, we find some $S(\mathbf{x}, \hat{\mathbf{y}})$ that is a better solution than $\hat{\mathbf{y}}$ with respect to the reward function. Therefore, we have to train the SPEN model such that, conditioning on $\mathbf{x}$, gradient-descent inference returns $S(\mathbf{x}, \hat{\mathbf{y}})$, thus guiding the model toward predicting a better output at each step. Figure 1 depicts an example of such a scenario.

For the gradient-descent inference to find $\hat{\mathbf{y}}_n = S(\mathbf{x}, \hat{\mathbf{y}})$, the energy of $(\mathbf{x}, \hat{\mathbf{y}}_n)$ must be lower than the energy of $(\mathbf{x}, \hat{\mathbf{y}})$ by margin $M$. We define the margin using scaled difference of their rewards:

$$M(\mathbf{x}, \hat{\mathbf{y}}, \hat{\mathbf{y}}_n)) = \alpha(R(\mathbf{x}, \hat{\mathbf{y}}_n) - R(\mathbf{x}, \hat{\mathbf{y}})), \tag{2}$$

where $\alpha > 1$ is a task-dependent scalar.

Now, we define at most one constraint for each training example $\mathbf{x}$:

$$\xi_{\mathbf{w}}(\mathbf{x}) = M(\mathbf{x}, \hat{\mathbf{y}}, \hat{\mathbf{y}}_n)) - E_{\mathbf{w}}(\mathbf{x}, \hat{\mathbf{y}}) + E_{\mathbf{w}}(\mathbf{x}, \hat{\mathbf{y}}_n) \leq 0 \tag{3}$$

As a result, our objective is to minimize the magnitude of violations regularized by $L_2$ norm:

$$\min_{\mathbf{w}} \sum_{\mathbf{x} \in \mathcal{D}} \max(\xi_{\mathbf{w}}(\mathbf{x}), 0) + c||\mathbf{w}||^2, \tag{4}$$

where $c$ is the regularization hyper-parameter. Algorithm 1 shows the search-guided training.

---

**Algorithm 1** Search-guided training of SPENs

---

$\mathcal{D} \leftarrow$ unlabeled mini-batch of training data
$R(.,.) \leftarrow$ reward function
$E_{\mathbf{w}}(.,.) \leftarrow$ input SPEN
**repeat**
    $\mathcal{L} \leftarrow 0$
    **for** each $\mathbf{x}$ in $\mathcal{D}$ **do**
        $\hat{\mathbf{y}} \leftarrow$ sample from $E_{\mathbf{w}}(\mathbf{y}, \mathbf{x})$.
        $\hat{\mathbf{y}}_n \leftarrow S(\mathbf{x}, \hat{\mathbf{y}})$     //search in reward function $R$ starting from $\hat{\mathbf{y}}$
        $\xi_{\mathbf{w}}(\mathbf{x}) \leftarrow M(\mathbf{x}, \hat{\mathbf{y}}, \hat{\mathbf{y}}_n) - E_{\mathbf{w}}(\mathbf{x}, \hat{\mathbf{y}}) + E_{\mathbf{w}}(\mathbf{x}, \hat{\mathbf{y}}_n)$
        $\mathcal{L} \leftarrow \mathcal{L} + \max(\xi_{\mathbf{w}}(\mathbf{x}), 0)$
    **end for**
    $\mathcal{L} \leftarrow \mathcal{L} + c||\mathbf{w}||^2$
    $\mathbf{w} \leftarrow \mathbf{w} - \lambda \nabla_{\mathbf{w}} \mathcal{L}$     //$\lambda$ is learning rate
**until** convergence

---

## 4 Related Work

Peng et al. (2017) introduce maximum margin rewards networks (MMRNs) which also use the indirect supervision from reward functions for margin-based training. Our work has two main

advantages over MMRNs: first, MMRNs use search-based inference, while SPENs provide efficient gradient-descent inference. Search-based inference, such as beam-search, is more likely to find poor local optima structured output rather than the most likely one, especially when output space is very large. Second, SG-SPENs gradually train the energy function for outputting better prediction by contrasting the predicted output with a local improvement of the output found using search, while MMRNs use search-based inference twice: once for finding the global optimum, which may not be accessible, and next, for loss-augmented inference, so their method heavily depends on finding the best points using search, while SG-SPEN only requires search to find more accessible local improvements.

Learning to search (Chang et al., 2015) also explores learning from a reward function for structured prediction tasks where the output structure is a sequence. The training algorithm includes a roll-in and roll-out policy. It uses the so-far learned policy to fill in some steps, then randomly picks one action, and fills out the rest of the sequence with a roll-out policy that is a mix of a reference policy and the learned policy. Finally, it observes the reward of the whole sequence and constructs a cost-augmented tuple for the randomly selected action to train the policy network using a cost-sensitive classifier. In the absence of ground-truth labels, the reference policy can be replaced by a sub-optimal policy or the learned policy. In the latter case, the training algorithm reduces to reinforcement learning. Although it is possible to use search as the sub-optimal policy, we believe that in the absence of the ground-truth labels, our policy gradient baselines are a good representative of the algorithms in this category.

For some tasks, it is possible to define differentiable reward functions, so we can directly train the prediction model using end-to-end training. For example, Stewart and Ermon (2017) train a neural network using a reward function that guides the training based on physics of moving objects with a differentiable reward function. However, differentiable reward functions are rare, limiting their applicability in practice.

Generalized expectation (GE) (Mann & McCallum, 2010), posterior regularization (Ganchev et al., 2010) and constraint-driven learning (Chang et al., 2007), learning from measurements (Liang et al., 2009), have been introduced to learn from a set of constraints and labeled features. Recently, Hu et al. (2016) use posterior regularization to distill the human domain-knowledge described as first-order logic into neural networks. However, these methods cannot learn from the common case of black box reward functions, such as the ones that we use in our experiments below on citation field extraction and shape parsing.

Chang et al. (2010) define a companion problem for a structured prediction problem (e.g., if the part-of-speech tags are legitimate for the given input sentence or not) supposing the acquisition of annotated data for the companion problem is cheap. Jointly optimizing the original problem and the companion problem reduces the required number of annotated data for the original problem since the companion problem would restrict the feasible structured output space.

Finally, there exists a body of work using reward functions to train structured prediction models with reward functions defined as task-loss (Norouzi et al., 2016; Bahdanau et al., 2017; Ranzato et al., 2016), in which they access ground-truth labels to compute the task loss, pretraining the policy network, or training the critic. These approaches benefit from mixing strong supervision with the supervision from the reward function (task-loss), while reward functions for training SG-SPENs do not assume the accessibility of ground-truth labels. Moreover, when the action space is very large and the reward function includes plateaus, training policy networks without pretraining with supervised data is very difficult. Daumé et al. (2018) address the issue of sparse rewards by learning a decomposition of the reward signal, however, they still assume access to reference policy pre-trained on supervised data for the structured prediction problems. In Daumé et al. (2018), the reward function is also the task-loss. The SG-SPEN addresses these problems differently, first it effectively trains SPENs that provide joint-inference, thus it does not require partial rewards. Second, the randomized search can easily avoid the plateaus in the reward function, which is essential for learning at the early stages. Our policy gradients baselines are a strong representative of the reinforcement learning algorithms for structured prediction problems without any assumption about the ground-truth labels.

# 5 Experiments

We have conducted training of SPENs in three settings with different reward functions: 1) Multi-label classification with the reward function defined as $F_1$ score between predicted labels and target labels. 2) Citation-field extraction with a human-written reward function. 3) Shape parsing with a task-specific reward function. Except for the oracle reward function that we used for multi-label classification, our other reward functions of citation-field extraction and shape parsing do not have access to any labeled data. In none of our experiments the models have access to any labeled data (for comparison to fully-supervised models see Appendix A).

## 5.1 Multi-label Classification

We first evaluate the ability of search-guided training of SPENs, SG-SPEN, to learn from light supervision provided by truncated randomized search. We consider the task of multi-label classification on Bibtex dataset with 159 labels and 1839 input variables and Bookmarks dataset with 208 labels and 2150 input variables.

We define the reward function as the $F_1$ distance between the true label set and the predicted set at training time, and none of the methods have access to the true labels directly, which makes this scenario different from fully-supervised training.

We also trained R-SPEN (Rooshenas et al., 2018) and DVN (value-matching training of SPENs) (Gygli et al., 2017) with the same oracle reward function and energy function. In this case, DVN matches the energy value with the value of the reward function at different structured output points generated by the gradient-descent inference. Similar to SG-SPEN, R-SPEN and DVN do not have direct access to the ground-truth. In general, DVNs require access to ground-truth labels to generate adversarial examples that are located in a vicinity of ground-truth labels, and this restriction significantly hurts the performance of DVNs. In order to alleviate this problem, we also add Gaussian noise to gradient-descent inference in DVN, so it matches the energy of samples from the energy function with their reward values, giving it the means to better explore the energy function in the absence of ground-truth labels. See Appendix D for more details on this experimental setup.

Table 1.B shows the performance of SG-SPEN, R-SPEN, and DVN on this task. We observed that R-SPEN has difficulty finding violations (optimization constraints) as training progresses. This is attributable to the fact that R-SPEN only explores the regions of the reward function based on the samples from the gradient-descent trajectory on the energy function, so if the gradient-descent inference is confined within local regions, R-SPEN cannot generate informative constraints.

## 5.2 Citation Field Extraction

Citation field extraction is a structured prediction task in which the structured output is a sequence of tags such as Author, Editor, Title, and Date that distinguishes the segments of a citation text. We used the Cora citation dataset (Seymore et al., 1999) including 100 labeled examples as the validation set and another 100 labeled examples for the test set. We discard the labels of 300 examples in the training data and added another 700 unlabeled citation text acquired from the web to them.

The citation text, including the validation set, test set, and unlabeled data, have the maximum length of 118 tokens, which can be labeled with one of 13 possible tags. We fixed the length of input data by padding all citation text to the maximum citation length in the dataset. We report token-level accuracy measured on non-pad tokens.

Our knowledge-based reward function is equivalent to Rooshenas et al. (2018), which takes input citation text and predicated tags and evaluates the consistency of the prediction with about 50 given rules describing the human domain-knowledge about citation text.

We compare SG-SPEN with R-SPEN (Rooshenas et al., 2018), iterative beam search with random initialization, policy gradient methods (PG) (Williams, 1992), generalized expectation (GE) (Mann & McCallum, 2010), and MMRN (Peng et al., 2017). Appendix E includes a detailed description of baselines and hyper-parameters.

Table 1: The comparison of SG-SPEN and other baselines using A) token-level accuracy for the citation-field extraction task, B) $F_1$ score for multi-label classification task, and C) intersection over union (IOU) for the shape-parser task.

A) Citation-field extraction

| Method | Accuracy | Inference time (sec.) |
|---|---|---|
| GE | 37.3% | - |
| Iterative Beam Search | | |
| K=1 | 30.5% | 159 |
| K=2 | 35.7% | 850 |
| K=5 | 39.3% | 2,892 |
| K=10 | 39.0% | 6,654 |
| PG | | |
| EMA baseline | 54.5% | < 1 |
| Parametric baseline | 47.9% | < 1 |
| MMRN | 39.5% | < 1 |
| DVN | 29.6% | < 1 |
| R-SPEN | 48.3% | < 1 |
| SG-SPEN | **57.1**% | < 1 |

B) Multi-label classification

| Method | Bibtex | Bookmarks |
|---|---|---|
| DVN | 42.2 | 34.1 |
| R-SPEN | 40.1 | 30.6 |
| SG-SPEN | **44.0** | **38.4** |

C) Shape parsing

| Method | IOU | Inference time (sec.) |
|---|---|---|
| Iterative Beam Search | | |
| K=5 | 24.6% | 3,882 |
| K=10 | 30.0% | 15,537 |
| K=20 | 43.1% | 38,977 |
| Neural shape parser | 32.4% | < 1 |
| SG-SPEN | **56.3**% | < 1 |

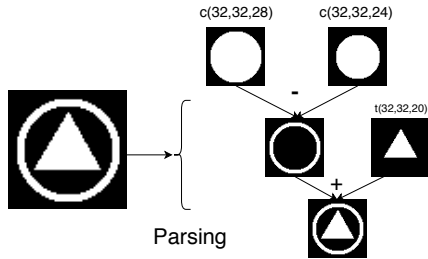

Figure 2: The input image (left) and the parse that generate the input input (right). The first two parameters of each shape shows its center location and the third parameter is its scale. A valid program sequence can be generated by post order traversal of the binary shape parse.

### 5.2.1 Results and Discussion

We reported the token-level accuracy of SG-SPEN and the other baselines in Table 1.A. SG-SPEN achieves highest performance in this task with 57.1% token-level accuracy. As we expect, R-SPEN accuracy is less than SG-SPEN as it introduces many irrelevant constraints into the optimization. Iterative beam search with beam size of ten gets about 39.0% accuracy, however, the inference time takes more than a minute per test example on a 10-core CPU. We noticed that using exhaustive search through a noisy and incomplete reward function may not improve the accuracy despite finding structured outputs with higher scores. DVN struggles in the presence of an inaccurate reward function since it tries to match the energy values with the reward values for the generated structured outputs by the gradient-descent inference. More importantly, DVNs learn best if they can evaluate the reward function on relaxed continuous structured outputs, which is not available for the human-written reward function in this scenario. MMRN also have problems to find the best path using greedy beam search because of local optima in the reward functions, but SG-SPEN and PG that are powered by randomized operations for exploring the reward function are more successful on this task.

### 5.2.2 Semi-Supervised Setting

We study the citation-field extraction task in the semi-supervised setting with 1000 unlabeled and 5, 10, and 50 labeled data points. SG-SPEN can be extended for the semi-supervised setting by using the ground-truth label instead of the output of the search whenever it is available. Similarly, for R-SPEN, we can evaluate the rank-based objective using a pair of model's prediction and ground

Table 2: Semi-supervised setting for the citation-field extraction task.

| No. | GE | PG | DVN | R-SPEN | SG-SPEN | SG-SPEN-sup | DVN-sup |
|---|---|---|---|---|---|---|---|
| 5 | 54.7 | 55.6 | 50.5 | 55.0 | **65.5** | 53.0 | 57.4 |
| 10 | 57.9 | 67.7 | 60.6 | 65.5 | **71.7** | 62.4 | 61.9 |
| 50 | 68.0 | 76.5 | 67.7 | 81.5 | **82.9** | 81.6 | 81.4 |

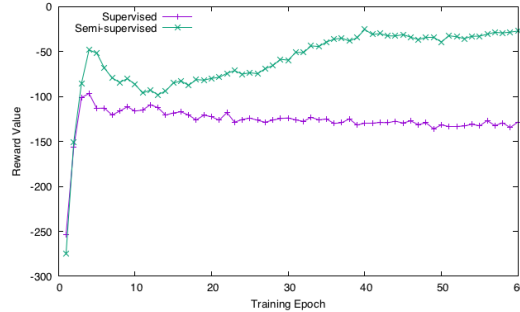

Figure 3: The test reward value of SG-SPEN's outputs trained in the supervised setting and semi-supervised settings with five labeled data points.

truth output when available. For DVNs, if the ground truth label is available, we use adversarial sampling as suggested by Gygli et al. (2017). We also reported the result of PG training with EMA baseline when the model is pre-trained with the labeled data. We reported the performance of GE based on Mann & McCallum (2010). We also reported the results of SG-SPENs when they are only trained with the labeled data using the citation reward function (SG-SPEN-sup).

Since the citation reward function is based on domain knowledge and is noisy, DVNs struggle in matching the energy values with the noisy rewards, so we also trained DVNs with token-level accuracy (not available for the unlabeled data) as the reward function (DVN-sup) for the reference.

SG-SPEN's performance is better than the other baselines in the presence of limited labeled data. However, since the training objective of R-SPEN and SG-SPEN are similar for the labeled data (both use rank-based objective), as we increase the number of labeled data, their performance become closer. DVNs also benefit from the labeled data, but it is very sensitive to noisy reward functions (see DVN and DVN-sup in Table 2). To better understand the behavior of SG-SPEN in the semi-supervised setting, we compare the reward value of test data for SG-SPENs during training with five labeled data in the fully-supervised and semi-supervised settings (see Figure 3). The unlabeled data helps SG-SPEN to better generalize to unseen data.

## 5.3 Shape Parsing

Shape parsing from computer graphics literature aims at parsing the input shape (2D image or 3D shape) into its structured elements as sequential instructions (program). These programs are in the form of binary operations applied on basic shape primitives (see Figure 2). However, for an input shape, predicting the program that can generate the input shape is a challenging task because of the combinatorially large output program space.

We apply our proposed SG-SPEN algorithm to the shape parsing task to show its superior performance in inducing programs for an input shape, without explicit supervision. Here we only consider the programs of length five, which includes two operations and three primitive shape objects: circle, triangle, and rectangle parameterized by their center and scale, which describes total 396 different shapes. Therefore, every program forms a sequence of five tags that each tag can take 399 possible values, including three operations and 396 shapes. The execution of a valid program results in $64 \times 64$ binary image (Figure 2).

For the shape parser task, we construct the reward function as the intersection over union (IOU) between a given input image and its constructed image from the predicted output program. This reward function is not differentiable as it requires executing the predicted program to generate the

final image. This is a difficult problem, first, the output space is very large, and second, many programs in the output space are invalid thus the reward function produces zero reward for them. We generated 2000 different image-program pairs based on Sharma et al. (2018), including 1400 training pair, 300 pairs for validation set, and 300 pairs for the test set. We dismiss the programs for the training data.

We compare SG-SPEN with R-SPEN, DVN, and iterative beam search with beam size five, ten, and twenty. We also apply neural shape parser proposed by Sharma et al. (2018) for learning from unlabeled data. See Appendix F for more details on this experiment.

### 5.3.1 Results and discussion

R-SPEN is not able to learn in this scenario because the samples from energy functions are often invalid programs and R-SPEN is incapable of producing informative optimization constraints. In other words, most of the pairs are invalid programs (with zero reward), thus having the same ranking with respect to the reward function, so they are not useful for updating the energy landscape to guide gradient-descent inference toward finding better predictions. DVN suffers from the same problem, without accessing to ground-truth data, the generated structured outputs by gradient-descent inference often represent invalid programs, and matching the value of invalid programs is not helpful toward shaping the energy landscape.

The results on this task are shown in Table 1.C (excluding the unsuccessful training of DVN and R-SPEN). SG-SPEN performs much better than neural shape parser because: first, the network is trained from scratch without any explicit supervision using policy gradients, which makes it difficult to find a valid program because of the large program space. Second, rewards are only provided at the end and there is no provision for intermediate rewards. In contrast, SG-SPEN makes use of the intermediate reward by searching for better program instructions that can increase IOU score. SG-SPEN quickly picks up informative constraints without explicit ground-truth program supervision (see Appendix C). The other advantage of SG-SPEN over neural shape parser in this task is its ability to encode long-range dependencies which enables it to learn the valid-program constraints quickly if the search operator reveals a valid program.

SG-SPEN also achieves higher performance compared to iterative beam search. Although in this scenario with an exact reward function, iterative beam search with higher beam sizes would gain better IOU, albeit with significantly longer inference time.

## 6 Conclusion

We introduce SG-SPEN to enable training of SPENs using supervision provided by reward functions, including human-written functions or complex non-differentiable pipelines. The key ingredients of our training algorithm are sampling from the energy function and then sampling from reward function through truncated randomized search, which are used to generate informative optimization constraints. These constraints gradually guide gradient-descent inference toward finding better prediction according to the reward function. We show that SG-SPEN trains models that achieve better performance compared to previous methods, such as learning from a reward function using policy gradient methods. Our method also enjoys a simpler training algorithm and rich representation over output variables. In addition, SG-SPEN facilitates using task-specific domain knowledge to reduce the search output space, which is critical for complex tasks with enormous output space. In future work we will explore the use of easily-expressed domain knowledge for further guiding search in lightly supervised learning.

### Acknowledgments

We would like to thank David Belanger, Michael Boratko, and other anonymous reviewers for their constructive comments and discussions.

This research was funded by DARPA grant FA8750-17-C-0106. The views and conclusions contained in this document are those of the authors and should not be interpreted as necessarily representing the official policies, either expressed or implied, of DARPA or the U.S. Government.

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
