[Supplementary Material · SG_SPEN_NeurIPS2019_supplementary.pdf]



Figure 4: The train set $F_1$ score for three different values of the reward margin $\delta$ in eq. 1 for the Bibtex multi-label classification task.

## A    Fully-Supervised Setting

For multi-label classification, DVNs achieve 44.7 $F_1$ score for Bibtex and 37.1 $F_1$ score for Book-marks, while SG-SPENs achieve 44.0 $F_1$ score for Bibtex and 38.4 $F_1$ score for Bookmarks. Since for this task, the reward function is the oracle $F_1$ score, the performance of SG-SPENs is on a par with the fully supervised setting on Bibtex and Bookmarks.

For citation-field extraction, we train SG-SPEN and DVN with token-level accuracy as the reward function on a training set of 300 labeled examples. SG-SPEN achieves 91.0% and DVN achieves 90.5% token-level accuracy. We also trained SG-SPEN with domain-knowledge based citation reward function, which resulted in 90.6% token-level accuracy.

For shape parsing, we trained the neural shape parser in the supervised setting as described in Sharma et al. (2018), which resulted in 60.0% intersection over union (IOU) comparing to 56.3% IOU of SG-SPEN without labeled data. Neural shape parser requires more labeled training data for better generalization.

## B    Selecting Reward Margin $\delta$

To show the importance of the reward margin in eq. 1, we train SG-SPEN for the Bibtex multi-label classification task with three reward margin values of 0.01, 0.002, 0.001. Figure 4 shows the train set $F_1$ score for the first 100 training epochs. SG-SPEN guided by search with the margin value of 0.001 is not able to learn, while the one with the margin of 0.002 struggles at the beginning of the training process as the search operator returns low reward output structures. SG-SPEN guided by search with the larger margin value of 0.01 has a better start. In more complex problems such as shape parsing, using a low reward margin can prevent the model from escaping low-reward regions. In general, using a larger margin and increasing the search budget increases the accuracy of the model as the search recovers better structures. Nevertheless, this higher accuracy is achieved at the price of an expensive search, which may significantly slow down the training.

## C    Search Budget and Informative Constraints

For the shape parser task, we gather the number of informative constrains (pairs with different reward rankings) of randomly selected batch of data at the first 1000 training steps (Figure 5, right). SG-SPEN can quickly pick up informative constraints even for this difficult task where the reward value of a notable portion of the search space is zero. We also observe that even at early stages of training the gradient-descent inference returns programs with positive rewards acknowledging that the SPEN rapidly learns to produce programs with valid structures.

We also collect the number search budget used by the search operator. We give the search budget of 100 to the search operator, which means it can randomly generate at most 100 structured outputs to find an improved structured output with respect to reward function (with the reward margin of 0.1)

Figure 5: Left) The average number of used search budget for each example in the first 1000 training iterations. Right) The number of informative constraints (pairs with different reward rankings) that search-guided training found for batches of 50 randomly selected training points in the first 1000 training steps. SG-SPEN generates at-most one informative constraint for each example.

Figure 6: Left: Number of optimization constrains of R-SPEN vs SG-SPEN. Right: Train-set $F_1$ score of R-SPEN, SG-SPEN, and DVN.

and the output of gradient-descent inference (eq. 1). As it is shown in Figure 5, left, the number of explored structured outputs by the search operator is relatively very small considering the output space ($399^5$). At the very beginning of the training process, the structured output generated by gradient-descent inference are very poor (mostly invalid programs), therefore, the search operator is not successful in finding an improved structured output using the given search budget. However, as soon as the search operator could find a valid program to guide SG-SPEN, the gradient-descent inference starts predicting more valid programs, so search operator becomes more successful in finding improved structured outputs without using the whole search budget.

# D    Multi-Label Classification

For the multi-label classification tasks we decompose the energy function into local energy and global energy as suggested by Belanger & McCallum (2016). For the feature network in the local energy term, we used 2-layer multi-layer perceptron (MLP) with 1000 hidden units with ReLU activation function for the Bibtex dataset, and used 3-layer MLP with two layers of 1000 hidden units with ReLU activation function for the Bookmarks dataset. We defined the global energy using 2-layer MLP over output variables with 15 and 50 hidden units with SoftPlus activation functions for Bibtex and Bookmarks, respectively. For SG-SPEN we used reward margin of 0.01, and tuned the number of inference iteration from $\{10, 15, 25, 30\}$, $\eta$ from $\{0.1, 0.5, 1.0, 2.0\}$, and $\alpha$ from $\{1, 10, 100\}$ using the performance of models on validation set. For this setting we set noise scale as $\sigma = 2\eta$. Bibtex and Bookmarks datasets do not have standard validation sets, so we randomly select 20% of training data as a fixed validation set for all the training models.

## D.1    Detailed Comparison

To better explore the behavior of R-SPEN vs SG-SPEN, we look at the number of informative constraints that each algorithm uses for training with the batch size of 100 examples and 10 inference iterations during the first 5000 iteration of training. R-SPEN generates one potential constraint for

every consecutive pairs (at most nine constraints for 10 iteration), while SG-SPEN generates only one. However, a fraction of these constraints violate the margin (eq. 3). Figure 6, left, shows the number of these constraints for R-SPEN and SG-SPEN. We also collect the train-set $F_1$ score of the same run for both algorithms as well as for DVN (Figure 6, right). SG-SPEN converges much faster than R-SPEN and DVN while using a lower but more informative amount of optimization constraints.

# E Citation Field Extraction

Figure 7: The parameterization of energy function using for citation-field extraction.

## E.1 Methods

**SG-SPEN**: We define the energy network using convolution neural networks over both word representation of input tokens and output tag distributions as shown in Figure 7. We use pretrained Glove vector representation with dimension of 50 for all the baselines,[1] however, we update word representations during the training.

**R-SPEN**: We use exactly the same energy function as SG-SPEN. The main difference between R-SPEN and SG-SPEN is their training algorithm.

**DVN**: Similar to R-SPEN, for DVN, we use exactly the same energy function as SG-SPEN. Also we find that DVNs learn better in our setting by optimizing the mean squared loss: $\|E(\mathbf{y}, \mathbf{x}) - \alpha R(\mathbf{y}, \mathbf{x})\|_2^2$.

**GE** uses human-written soft-constraints as labeled features to constrain the model's prediction with respect to unlabeled data. For GE, we include the results from Mann & McCallum (2010) for the same setting, for which they have used the same test set and 1000 unlabeled training data.

**Iterative Beam Search**: We started from a random tag sequence, and then iteratively run beam search with beam size of $K$ until the top $K$ sequences remains the same within ten iterations. We re-run this iterative beam search with ten random restarts and reports the accuracy of the sequence with the highest score.

**PG**: We also train a recurrent neural network (RNN) using policy gradient methods. For each word in the input sequence, the model will predict the output tag given the last hidden states of RNNs, last predicted tag and current input. The rewards are the value of our human-knowledge score function over the input token sequence and predicted output of RNNs. To reduce the variance of gradients, we use two different baseline models: exponential moving average (EMA) baseline and parametric baseline. EMA defines the baseline as weighted average over history rewards and the current reward: $B_t = B \leftarrow \alpha B + (1 - \gamma)r$, where $r$ is the average reward of the current batch and $\gamma$ is the decaying rate. For the parametric baseline, we use the current token $x_t$, previous hidden state $h_{t-1}$, and output $y_{t-1}$ from RNN to predict the baseline using linear regression: $B_t(x_t, h_{t-1}, y_{t-1}) = W[h_{t-1}; x_t; y_{t-1}] + b$, where $W$ and $b$ are the parameters of the baseline learned by minimizing the mean square distance between the baseline and reward. During training, we found that the probability distribution produced by policy function $\pi_\theta$ tends to polarize before the model becomes optimal. To

Figure 8: The parameterization of energy function for shape parsing. The network has two parts: first takes the probability distribution over the output program and outputs a fixed dimension embedding, and the second part takes the binary images as input, which is convolved to give fixed length embedding. The two embeddings are concatenated and passed through an MLP to output energy function.

maintain the exploration ability of the model, we add entropy regularization in our object function. In our experiments, we also attempted to re-normalize the probability of sampled sequences, but since it did not show better performances in this dataset, we exclude it in our final PG models.

**MMRN** has the same architecture as PG, but trained with the max-margin objective (Peng et al., 2017).

### E.2 Hyper-Parameter Tuning

We select the hyper-parameters using grid search and based on the performance of models on the validation set.

For DVN, R-SPEN, and SG-SPEN, we tuned $\eta$ from $\{0.1, 0.5, 1.0, 2.0\}$, the number of iteration = $\{15, 20, 25, 30, 35\}$, as well as the number of filters of text cnn = $\{64, 128, 256\}$. For SG-SPEN, we also explored $\alpha = \{1, 10, 100\}$, and interestingly larger values of $\alpha$ was preferred based on the performance on validation set.

For PG and MMRN baselines, we used beam size of 10, and tuned the dimension of hidden layers = $\{10, 30, 50\}$, learning rate = $\{0.01, 0.001\}$, batch size = $\{16, 128, 512\}$. In addition, for PG + EMA where baseline is $B_t = B \leftarrow \alpha B + (1 - \alpha)r$, we chose $\alpha = \{0.5, 0.7, 0.9\}$. We also tuned the weight of entropy gradient of PG from $\{0.1, 1.0\}$.

## F Shape Parser

Neural shape parser includes an encoder and a decoder. The encoder consists of 2d convolution and max pooling layers with ReLU non-linearity, that takes an image as input and gives a fixed dimensional feature vector as output. The decoder is a GRU, that takes the image features as input at every time step. The hidden state of the GRU at every time step is transformed by two fully connected layers and a softmax layer to output a distribution over program instructions. we select the number of GRUs' hidden states from $\{512, 1024, 2028\}$, dropout rate from $\{0.2, 0.4, 0.6, 0.8\}$, and learning rate from $\{0.01, 0.05, 0.001\}$.

The training is done using policy gradient algorithm with running average baseline with gamma of 0.9 and mini batch of 64 images. We use stochastic gradient descent with 0.9 momentum. For R-SPEN, DVN, and SG-SPEN, we use the energy architecture of Figure 8, and tuned $\eta$ from $\{0.1, 0.5, 1.0, 2.0\}$ and inference iteration from $\{15, 20, 25, 30\}$. We use $\delta = 0.1$, $\sigma = 2\eta$, and $\alpha = 100$.

## Footnotes

[1]https://nlp.stanford.edu/projects/glove/