[Reviews · NeurIPS 2019]

Reviewer 1



Post-feedback update: Thank you for your update. Your additional results will strengthen this paper, and I still think it should be accepted. ------------------------------------------------------------------------------------------------------------- Originality: The ideas presented in this paper represent somewhat of a synthesis of other ideas. Specifically, it combines the basic overall framework for SPEN training using a reward signal introduced by [1] with the idea of adding in random search to find reward scoring violations, which has been used in the past by various papers (which are cited appropriately in this work). However, this exact combination is novel. Quality: The motivation behind using random search to augment the generation of labels to use for training the model is sound and verified empirically. Numerous appropriate baselines are included, ranging from beam search-type approaches to more directly comparable approaches such as [1], and the introduced approach outperforms all of them. There are additional results presented that further reinforce some of the ideas and motivations introduced when describing the model: specifically, that some problems use reward signals that are somewhat uninformative in many regions of the search space, and that using solely gradient-based approaches to find new training points can cause the model to get "stuck" in local optima. It would have been interesting to see experiments in a semi-supervised setting to see how much this approach can augment training using a limited amount of fully-labeled training data, but the content in this paper is sufficient to be interesting on its own. Clarity: The ideas are presented clearly and logically, and there are no problems in understanding the problem, the motivations behind the solution, and how the solution addresses shortcomings of other approaches. The experiments are described in adequate detail and the results are easy to understand. Significance: The ability to utilize a reward function for training instead of full supervision is appealing, since getting full training labels can be much more expensive than being able to provide a reward function. The presented results indicate that this approach can provide significant improvements over competing approaches that do not use full supervision and thus is worthwhile to use. [1]Rooshenas, A., Kamath, A., and McCallum, A. Training structured prediction energy networks with indirect supervision. NAACL: HLT, 2018.

Reviewer 2



===== Update following rebuttal ==== The additional experiments strengthen the submission so I am updating my score to 6. I think there should be a more serious discussion of the accuracy-vs-computation tradeoff for the truncated randomized search (number of steps, margin requirement, etc). Unfortunately, this point was not addressed in the rebuttal. ===== Overview: This paper proposes an approach called Search-Guided SPENs for training SPENs from reward functions defined over a structured output space. The main idea is to refine the prediction of the energy network by searching for another output that improves its reward. If the search is not too expensive, this can speedup training and improve performance. Rather than gradient-based search (previously suggested in R-SPENs), which can get stuck, a truncated randomized search with reward values is proposed here. Experiments suggest that the proposed approach can indeed improve performance on several tasks with relatively cheap computation. Overall, the approach is presented clearly and seems to improve over previous work in experiments. However, I feel that some aspects which are left as future work, such as results in the fully-/semi-supervised settings and investigation on the effect of the search procedure, should actually be included in this work. Detailed comments: The effectiveness of the randomized search procedure seems central to this approach, but it is discussed in Appendix B. It seems like this should get more attention and be included in the main text. Also, it seems interesting to explore smarter search procedures that exploit domain knowledge and compare them to the randomized one in terms of the computation-accuracy trade-off. This is mentioned as future work, but feels central to understanding the merits of the proposed approach. Experiments: * In the experiments, it is interesting to add a comparison to a fully supervised baseline that does use ground-truth labels (e.g., vanilla SPENs) in order to get a sense of performance gaps with reward-based training, whenever ground-truth labels are available. In multilabel classification this is especially relevant since the reward function actually depends on the true labels. * An interesting question is whether the proposed approach can improve over supervised training (e.g., vanilla SPENs) in fully-supervised and semi-supervised settings, and not just learn a predefined reward function, but this is not addressed in the paper. * How is the trade-off parameter alpha between energy and reward in eq (2) chosen? Presumably some tuning is required for this hyperparameter which should be considered for the training time comparison. * Are the results reported wrt to some ground-truth or wrt the reward function? I was expecting to see both in Table 1. This distinction can help understand the source of errors (optimization vs mismatch of reward and performance measure). Minor: Line 120: notice that y_n may not exist if y_s is already near optimal. This is handled later (line 135), but only as failure of the search procedure, and not because of near-optimality. Would be good to clarify. Line 229: “thus we this” Line 301: “an invalid programs”

Reviewer 3



============= Update after rebuttal ================ I have read the other reviews and the authors's rebuttal. I appreciate the additional experiments presented by the authors and will be upgrading my recommendation to a 7 to reflect this. I still think more of these additional experiments and larger-scale experiments would increase the paper's significance by a lot, but I think it does pass the bar for publication in its current state. ============================================= The paper introduces a new twist on the ranked-based approach to training structured prediction energy networks via light supervision (where light supervision means that the learning signal for the energy function learned by the model comes from enforcing that the energy levels are consistent with the levels of a reward function). Instead of picking random samples guided by the energy function, which will often offer the same reward (since this function is mostly uninformative and has wide plateaus), the samples are sampled first through gradient-based inference, and then via local search on the reward function itself, to make sure that there is a difference in value between the two samples. The algorithm is run on 3 small scale structured prediction datasets and is shown to outperform the previous ranked-based SPEN training algorithm. Originality The main contribution (i.e. the new sampling of the datapoints) is a new twist on an existing algorithm. As such it's not very original, though novel. The related work is extensively cited and the delineation with the contributions of the paper is well done. Clarity The paper is very well written and easy to read. While probably a bit verbose, it explain in details both the SPEN models, their training, the new proposed training and its relations with the previous work. There are a couple of surprising claims though, which are worth noting (see details below). Quality While the algorithm is applied on 3 tasks, all of them are very small scale. One cannot really evaluate the promise of the approach on such datasets, so in this sense the paper might not be quite ready yet. Otherwise the paper is technically sound. Significance Again, while the innovation is fairly minor, it might result in big improvements empirically, but one cannot readily verify it for the lack of large scale task in the experimental section. Question: This is beyond the scope of the paper (although it would make for a nice addition and help strengthen its originality. In the setup considered, we only use the light supervision of the reward function. On a fully labeled dataset, it is possible to compute many rewards based on the ground truth labels. Would you expect that training with both the supervised loss as well as the lighter supervision of these additional rewards would work better than to train simply with the supervised loss? All told, this paper is on the fence as regards acceptance. It is very clear and of good quality, but might still be improved with larger scale experiments. Details l157-159: the claim is a bit surprising, considering gradient descent is notoriously prone to converging to poor stationary points as opposed to the global optimum. l200 & l202: the citation should read Daumé and not Daumé III. The way to do it in a bib file is the following: author = {Daum\'e, III, Hal and ...} l215, the claim that the models do not have access to the ground truth is misleading at least in the case of multi-label classification where the reward is a direct function of the labels.

[Author Response · NeurIPS 2019]

Table 1: Semi-supervised setting for the citation-field extraction task.

| No. | GE | PG | DVN | R-SPEN | SG-SPEN | SG-SPEN-sup | DVN-sup |
|-----|------|------|------|--------|---------|-------------|---------|
| 5 | 54.7 | 55.6 | 50.5 | 55.0 | **65.5** | 53.0 | 57.4 |
| 10 | 57.9 | 67.7 | 60.6 | 65.5 | **71.7** | 62.4 | 61.9 |
| 50 | 68.0 | 76.5 | 67.7 | 81.5 | **82.9** | 81.6 | 81.4 |

Figure 1: The test reward value of SG-SPEN's outputs trained in the supervised setting and semi-supervised settings with five labeled data points.

We appreciate the reviewers' comments and concerns. Two common questions of reviewers are comparisons to fully-supervised and semi-supervised settings that we address here. The reported results here and in the main paper are with respect to labeled test data on the reported task-specific measure, which is the same as the reward function for shape parsing and multi-label classification. For citation-field extraction, the reward function and accuracy measure are different. The citation reward function is based on domain knowledge and is noisy, and the ground-truth label may not have the highest reward value.

**Fully-supervised setting**: For multi-label classification, DVNs achieve 44.7 $F_1$ score for Bibtex and 37.1 $F_1$ score for Bookmarks, while SG-SPENs achieve 44.0 $F_1$ score for Bibtex and 38.4 $F_1$ score for Bookmarks. Since for this task, the reward function is the oracle $F_1$ score, the performance of SG-SPENs is on a par with the fully supervised setting on Bibtex and Bookmarks. For citation-field extraction, we train SG-SPEN and DVN with token-level accuracy as the reward function. SG-SPEN achieves 91.0% and DVN achieves 90.5% token-level accuracy. We also trained SG-SPEN with domain-knowledge based citation reward function, which resulted in 90.6% token-level accuracy. For shape parsing, we trained the neural shape parser in the supervised setting as described in Sharma et al. (2018), which resulted in 60.0% intersection over union (IOU) comparing to 56.3% IOU of SG-SPEN without labeled data. Neural shape parser requires more labeled training data for better generalization.

**Semi-supervised setting**: We study the citation-field extraction task in the semi-supervised setting with 1000 unlabeled and 5, 10, and 50 labeled data points. SG-SPEN can be extended for the semi-supervised setting by using the ground-truth label instead of the output of the search whenever it is available. Similarly, for R-SPEN, we can evaluate the rank-based objective using a pair of model's prediction and ground truth output when available. For DVNs, if the ground truth label is available, we use adversarial sampling as suggested by Gygli et al. (2017). We also reported the result of PG training with EMA baseline when the model is pre-trained with the labeled data. We reported the performance of GE based on Mann & McCallum (2010). We also reported the results of SG-SPENs when they are only trained with the labeled data using the citation reward function (SG-SPEN-sup). Since the citation reward function is based on domain knowledge and is noisy, DVNs struggle in matching the energy values with the noisy rewards, so we also trained DVNs with token-level accuracy (not available for the unlabeled data) as the reward function (DVN-sup) for the reference.

SG-SPEN's performance is better than the other baselines in the presence of limited labeled data. However, since the training objective of R-SPEN and SG-SPEN are similar for the labeled data (both use rank-based objective), as we increase the number of labeled data, their performance become closer. DVNs also benefit from the labeled data, but it is very sensitive to noisy reward functions (see DVN and DVN-sup in Table 1). To better understand the behavior of SG-SPEN in the semi-supervised setting, we compare the reward value of test data for SG-SPENs during training with five labeled data in the fully-supervised and semi-supervised settings (see Figure 1). The unlabeled data helps SG-SPEN to better generalize to unseen data.

[Meta-Review · NeurIPS 2019]

All the reviewers support this paper after the authors' response clarified the raised issues. I recommend the authors to update the paper according to the suggestions.